# Efficient Recovery of Cu from Wasted CPU Sockets by Slurry Electrolysis

**Wenjing Chen [1], Manning Li [2] and Jiancheng Tang [1,3,*]**

[1] School of Physics and Materials Science, Nanchang University, Nanchang 330031, China
[2] School of Advanced Manufacturing, Nanchang University, Nanchang 330031, China
[3] International Institute for Materials Innovation, Nanchang University, Nanchang 330031, China
[*] Correspondence: tangjiancheng@ncu.edu.cn; Tel.: +86-83969559

**Abstract:** In order to maximize the reuse of used electronic component resources, while reducing environmental pollution, Cu metal was recycled from wasted CPU sockets by the reformative slurry electrolysis method. However, the influences on the regulation of the Cu recovery rate and purity from waste CPU slots, by slurry electrolysis, has not been systematically elucidated in previous studies. In this work, the effects of $H_2SO_4$ concentration, slurry density, $NH_4Cl$ concentration, current density, and reaction time, on the recovery rate and purity of Cu in slurry electrolysis, were researched by systematic experimental methods. The results showed that the recycled metal elements were mainly present as powders from the cathode, rather than in the electrolyte. Moreover, the metallic elements in the cathode powder consisted of mostly Cu and small amounts of Sn and Ni. The recovery rate and purity of Cu were up to 96.19% and 93.72%, respectively, with the optimum conditions being: an $H_2SO_4$ concentration of 2 mol/L, slurry density of 30 g/L, $NH_4Cl$ concentration of 90 g/L, current density of 80 mA/cm$^2$, and reaction time of 7 h. Compared with previous studies, the Cu recovered in this experiment was present in the cathode powder, which was more convenient for the subsequent processing. Meanwhile, the recovery rate of Cu was effectively improved. This is an important guideline for the subsequent application of slurry electrolysis for Cu recovery.

**Keywords:** wasted CPU socket; slurry electrolysis; recovery rate of Cu; purity of Cu; current density; slurry density

## 1. Introduction

Today, environmental protection is an urgent topic. With the rapid development of society, resources are rapidly being consumed and a large amount of garbage is generated, most of which is not fully used, or could be reused [1,2]. Among all waste, the growth rate of e-waste has accelerated with the frequency of technological updates [3–5]. In 2016, the global output of e-waste was 44.7 million tonnes, while in 2019, the figure reached 53.6 million tons, with an average annual growth rate of 6.64 percent. Asia alone generated 24.9 million tonnes of e-waste. The global production of e-waste is expected to exceed 70 million tonnes by 2028 [6].

The composition of e-waste is complex, and up to 69 elements have been found in it, including various precious metals (gold, copper, platinum, and palladium) [7,8]. Besides that, there are also various pollutants (e.g. plastics) found in e-waste [9]. If not handled properly, it can cause serious harm to the environment and human health. At the same time, e-waste, also known as "misplaced resources", generates a lot of economic losses [10,11]. The potential value of the e-waste generated in 2019, was estimated to be 57 billion US dollars [6]. Research has shown, that the Cu content in the printed circuit board (PCB) of an ordinary notebook computer is about 20 wt.%, which is much higher than the global average copper grade [12,13].

Currently, the main recycling technologies for e-waste are mechanical-physical treatment technology, biometallurgical recovery technology, and new recycling processes, such

as supercritical fluid recycling treatment technology and slurry electrolysis recycling treatment technology [14–16]. Slurry electrolysis is a technology that simultaneously performs leaching and electrodeposition. The reaction of metals in slurry electrolysis is divided into two parts: anode leaching and cathode reduction.

CPU sockets need to be pre-processed before slurry electrolysis. Zhao et al. [17] separated the valuable metals from wasted mobile phone PCB particles using the liquid–solid fluidization technique. Hanafi et al. [18] reported that a ball-milling machine was preferable to a disc-milling machine, due to its uniform pulverization, but ball-milling required a longer time than disc-milling. Yi et al. [19] cut CPU sockets first, and then shredded them into pieces, with diameters less than 2 mm, using a cutting mill.

Electrolysis experiments are responsible for the recovery of Cu from e-waste. Veit et al. [20] recovered metallic copper from wasted printed wiring boards by acid leaching, and then electrolytic deposition of the acid leaching solution was carried out, which eventually obtained metallic copper powder, with a purity of up to 98%. Subsequent researchers have further explored methods for the efficient recovery of Cu by changing different variables. Chu et al. [21] used electrolysis to recover copper powder from wasted PCBs and studied the effect of different factors on current efficiency and copper powder particles. Min et al. [22] developed a multi-oxidation coupling with electrolysis strategy, for purifying PCB wastewater and recovering copper. Guimarães et al. [23] found that electrolyte stirring and temperature increase favor the cathodic recovery of copper from PCB powder concentrates, by direct electroleaching. Byung et al. [24] simultaneously extracted precious metals such as gold, palladium, and platinum, from scrap (PCBs) and cellular auto catalysts, by smelting. Zhao et al. [25] developed an efficient process based on a hammer mill, pneumatic column separator, and electrostatic separator, for recovering copper from scrap PCBs. Zhang et al. [3] used the slurry electrolysis method to study the ultrafine copper powder obtained by adding different additives. Wang et al. [26] performed electrolysis in a centrifuge, to recover high purity copper powder from a polymetallic solution. The purity, current efficiency, and recovery of copper on the centrifuged electrode, were significantly improved due to enhanced mass transfer compared to the non-rotating electrode. Liu et al. [27] investigated the recovery and purification of Pd from waste multilayer ceramic capacitors (MLCCs) using electrodeposition, and proposed an efficient and environmentally friendly process for the recovery of waste MLCCs. Cocchiara et al. [28] explored the electrochemical recovery of Cu by cyclic voltammetry, the results showed that $H_2SO_4$–$CuSO_4$–NaCl could efficiently leach Cu from wasted PCBs, so that electronic components can be more easily disassembled in their undamaged state, allowing for effective recycling and valorization of base materials.

At present, the object of metal recycling is mostly wasted printed circuit boards (PCBs). The composition of PCBs is affected by the manufacturer, age, and origin, but Cu is always one of the most abundant metals in these materials [29]. CPU sockets act as important components, utilized to connect CPUs and motherboards. To reduce resistance, the content of Cu in CPU sockets is relatively high [30]. Although a wide range of studies have focused on the recovery of precious metals from e-waste, researchers have mainly focused on the extraction of metals in the electrolyte. This experiment investigates the extraction of metals at the cathode, as well as the enhancement of the Cu recovery rate and purity.

## 2. Materials and Methods

### 2.1. Materials

The reagents used in the study included nitric acid ($HNO_3$, 65–68%), sulfuric acid ($H_2SO_4$, 98%), hydrochloric acid (HCl, 36%), ammonium chloride ($NH_4Cl$), and hydrogen peroxide ($H_2O_2$, 30 wt.%). All chemicals for the experiment used in the present study were analytically pure and purchased from Shanghai Aladdin Biochemical Technology Co., Ltd., Shanghai, China. All necessary leachate solutions used in the study were prepared from the materials mentioned above, in deionized water.

### 2.2. Waste Central Processing Unit Socket

CPU sockets were the same model (LGA1366) obtained from wasted computers, as shown in Figure 1a. The samples of CPU sockets were manually shredded into approximately $10 \times 10$ mm$^2$ pieces and stored for further preparation, as shown in Figure 1b. To obtain the extremely tiny powder, the pieces were broken by a cutting mill (QE-300), as displayed in Figure 1c. To separate the metallic powder from non-metallic powder, the CPU socket powder was put in a centrifuge, which caused the layering of the metallic and non-metallic powders, and then the non-metallic powder was removed by mechanical vibration. After the sampling, the metallic powder was washed with ethanol and acetone, and finally dried at 70 °C for 2 h.

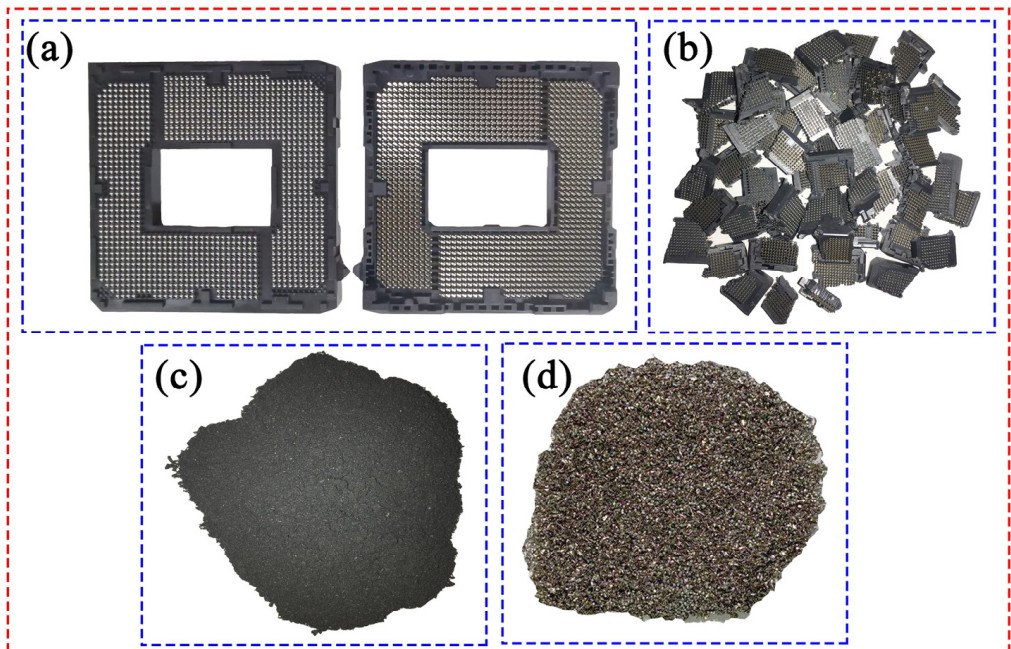

**Figure 1.** (**a**) CPU socket, (**b**) CPU socket pieces, (**c**) CPU socket powder, (**d**) metallic powder.

### 2.3. The Detection of Metallic Powder

X-ray diffraction (XRD) was used to detect the composition of metallic elements in the CPU sockets. To further probe the metal content in the CPU sockets, 0.1 g of collected CPU socket powder was put into a polytetrafluoroethylene (PTFE) crucible, and then 20 mL aqua regia (HNO$_3$:HCl = 1:3) was added. When the aqua regia was heated to 200 °C and then kept for 2 h, the lid of the crucible was opened to evaporate the solution to 1 mL. Finally, the remaining solution was fixed to 50 mL by adding deionized water, and stored. The leachate was analyzed by an inductively coupled plasma atomic emission spectrometer (ICP-AES, Optima 8000). Furthermore, the powder samples were also characterized via scanning electron microscopy (SEM, Quanta 200FEG), that was equipped with energy dispersive spectrometer mapping (EDS-Mapping).

### 2.4. Slurry Electrolysis Experiment

Figure 2a shows the instrument used in the slurry electrolysis experiment. The PTFE electrolysis cell was composed of a cathode part ($7 \times 6 \times 4$ cm$^3$) and anode part ($7 \times 6 \times 6$ cm$^3$). A graphite rod was utilized as the anode, while the cathode was titanium plate, and the electrodes were parallel to each other, with the distance kept constant. Electricity during the experiment was provided by a DC supplier (MS305DS, Dongguan Maihao Electronic Technology Co., Dongguan, China).

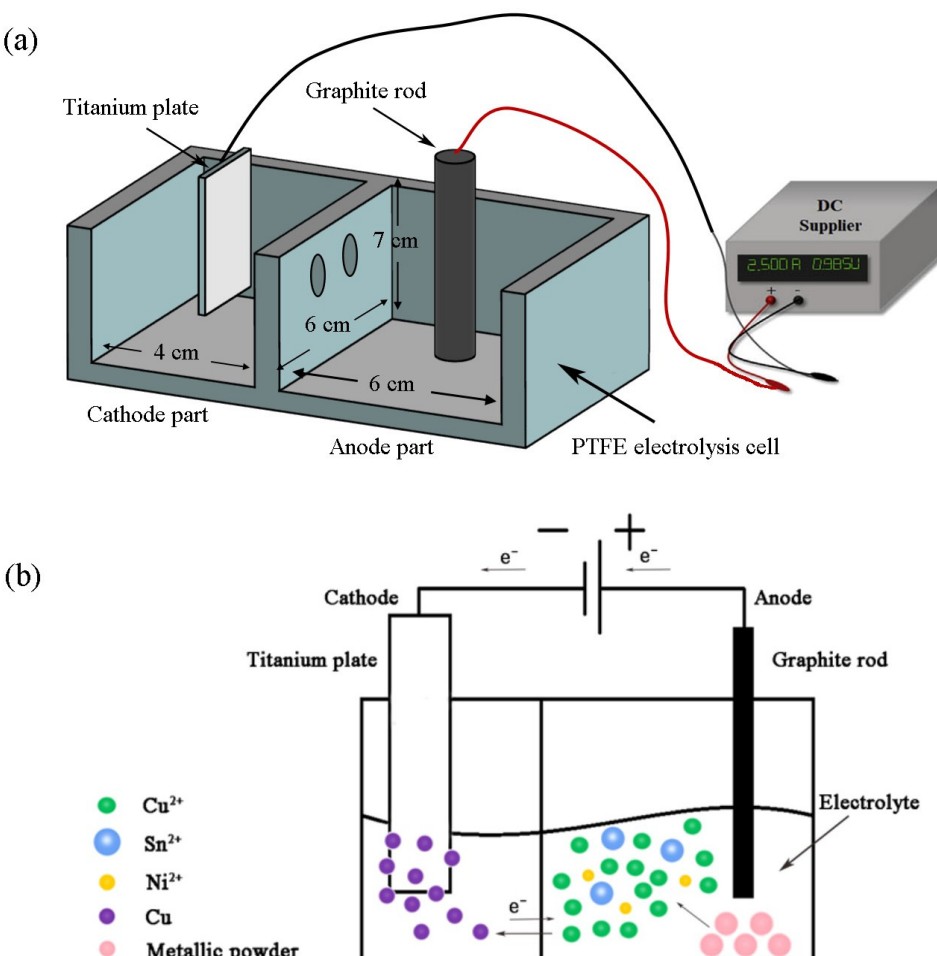

**Figure 2.** Schematic diagram of (**a**) slurry electrolysis experimental equipment, and (**b**) slurry electrolysis.

The CPU socket powder sample was put into the anode part, and 100 mL electrolyte, composed of 10 mL $H_2O_2$ (30 wt.%), 90 mL of different $H_2SO_4$ concentration solution, and varied weight of $NH_4Cl$ (3–9 g) was added. Furthermore, the rotor of the magnetic stirrer was set at 300 rpm in the anode section. Finally, a DC supplier was connected to the graphite rod and titanium plate, and energized for the experiments. In the initial electrolysis experiments, without $NH_4Cl$, the current density was set to 80 mA/cm$^2$. This is the optimal current density for such electrolysis experiments, based on the study of Zhang et al. [3]. The metal powder in the anode was dissolved by the combined effect of the current and $H_2SO_4$.

To explore the influence of slurry density, the concentration of $H_2SO_4$ solution, $NH_4Cl$ concentration, current density, and reaction time on the slurry electrolysis experiment, the slurry density was varied between 30, 50, and 70 g/L, the concentration of the $H_2SO_4$ solution was varied between 1, 2, 3, and 4 mol/L, the $NH_4Cl$ concentration was varied between 30, 60, and 90 g/L, the current density was varied between 40, 80, and 120 mA/cm$^2$, and the reaction time was varied between 3, 5, and 7 h.

After each experiment, ICP-AES was applied, to analyze the metal concentration of the powders obtained from the cathode and anode, and the cathode powder was further characterized by SEM and EDS. The relative error of the EDS analysis varies with the elemental content; the smaller the content, the larger the relative error. The relative error is 2% when the content is >20 wt.%, 10% when the content is between 3 wt.%–20 wt.%, 30% when the content is between 1 wt.%–3 wt.%, and 50% when the content is < 1 wt.%. The whole experimental process is depicted in Figure 3.

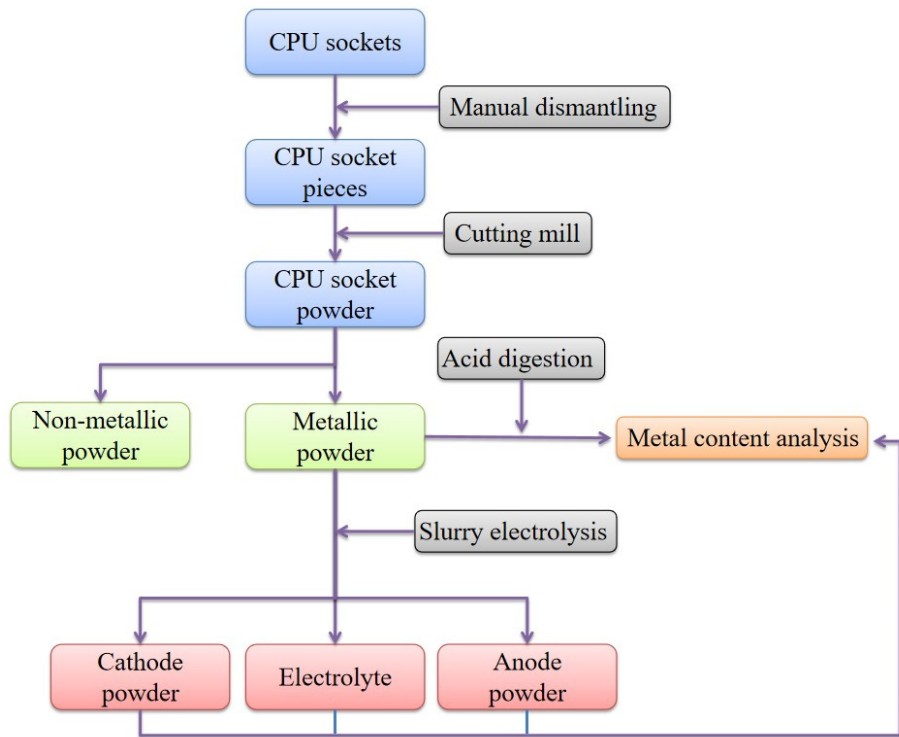

**Figure 3.** The procedure of the slurry electrolysis experiment.

*2.5. Characterization*

Recovery and separation of different metals from slurry electrolysis by electrodeposition are based on their different reduction potentials. The reactions in the cathode are shown by Equations (1)–(6) and the reactions in this experiment are shown in Figure 2b.

$$Cu^{2+} + 2e^- \rightarrow Cu \qquad E = 0.3419 \text{ V} \tag{1}$$

$$Sn^{2+} + 2e^- \rightarrow Sn \qquad E = -0.1375 \text{ V} \tag{2}$$

$$Ni^{2+} + 2e^- \rightarrow Ni \qquad E = -0.267 \text{ V} \tag{3}$$

$$2H^+ + 2e^- \rightarrow H_2 \qquad E = 0 \text{ V} \tag{4}$$

where E is the electrode potential, which is a value relative to the potential of the standard hydrogen electrode (SHE).

Metal recovery rate (*r*) and purity (*P*) are calculated by following Equations (5) and (6):

$$rij = \frac{mij}{M} \times 100\% \tag{5}$$

where *i* is Cu, Sn, or Ni; *j* is the electrolyte and cathode; $r_{ij}$ is the recovery rate of *i* in *j* (%); $m_{ij}$ is the mass of *i* obtained in *j* (g); and *M* is the mass of metal contained in the waste CPU socket (g).

$$Pij = \frac{Mij}{Mj} \times 100\% \tag{6}$$

where *i* is Cu, Sn, or Ni; *j* is the anode, electrolyte, and cathode; $P_{ij}$ is the purity of metal of *i* in *j* (%); $M_{ij}$ is the mass of *i* obtained in *j* (g); and $M_j$ is the mass of metal obtained in *j* (g).

## 3. Results and Discussion

*3.1. Characterization of the CPU Socket Powder Sample*

As shown in Figure 4a, the main components of the CPU socket powder are pure metals, including Cu, Ni, and Sn, also, the peaks of Ag and Au were discovered.

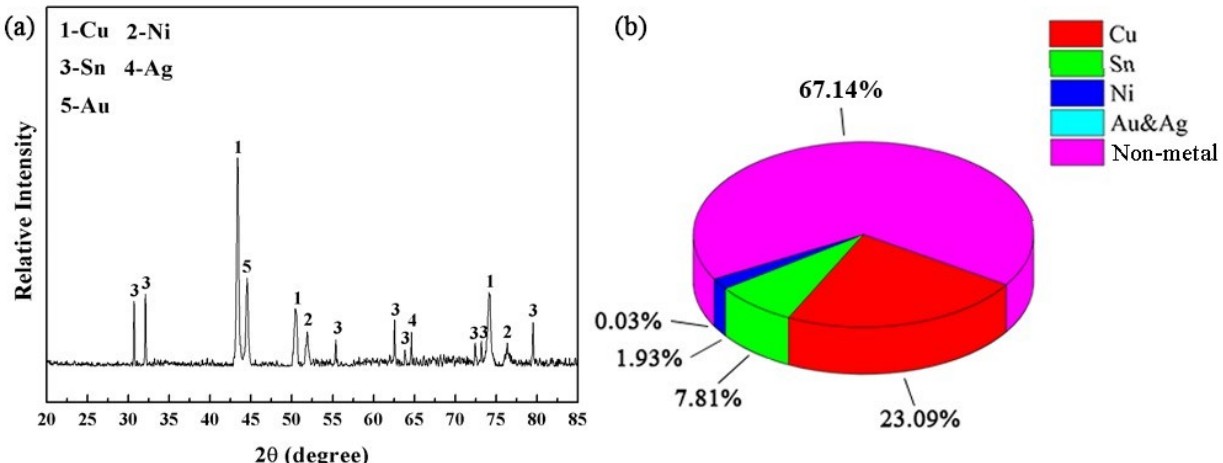

**Figure 4.** (**a**) XRD of the CPU socket metal powder, (**b**) compositions of the CPU socket powder.

The metal content of the metallic powder was detected by ICP-AES, which was consistent with the composition analysis by XRD. The amounts of different components in the CPU socket sample are summarized in Figure 4b.

Figure 5(a-1) shows an SEM image of the CPU socket powder fracture surface. It was detected, by EDS mapping, that the main elements of the CPU socket powder fracture surface were Cu and Sn. The results are displayed in Figure 5(a-2,a-3). Figure 5(b-1) shows an SEM image of the CPU socket powder surface, and Figure 5(b-2) shows the EDS mapping of the SEM image. It was indicated that only Ni was detected. In Figure 5(c-1), part 1 is the CPU socket powder surface and part 2 is the CPU socket powder fracture surface, Figure 5((c-2)–(c-4)) shows that the metal matrix is Cu and Sn, and Ni is plated on the surface as a coating.

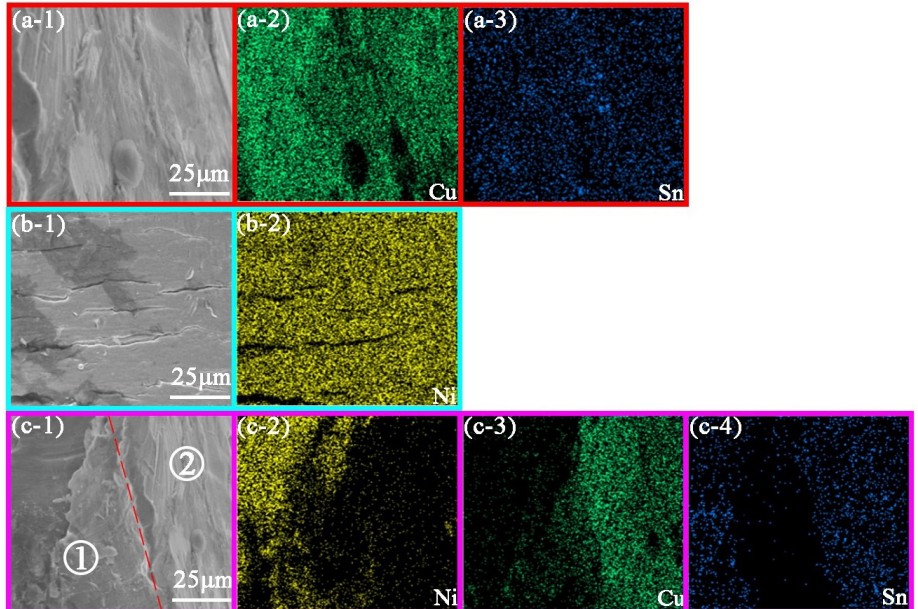

**Figure 5.** ((**a-1**,**b-1**,**c-1**)) SEM images of different parts of the CPU socket powder, (**a-2**,**a-3**,**b-2**,**c-2**–**c-4**) element mapping of Cu, Ni, and Sn for ((**a-1**,**b-1**,**c-1**)).

### 3.2. Characterization of Cathode Powder

A piece of metallic foil covering the titanium plate, and a large amount of powder adhering to the metallic foil, are shown in Figure 6a. They were obtained under the following experimental conditions: slurry concentration of 30 g/L, $H_2SO_4$ concentration

2 mol/L, current density 80 mA/cm$^2$, reaction time 7 h, and reaction temperature 15 °C. Based on the color of the metal, it can be deduced that the percentage of Cu in the metal was high. Metal foil was achieved after removing the powder from the titanium plate, and Figure 6b shows an SEM analysis of the side of the metal foil near the titanium plate. It is clear that a layer of dense metal, whose dominant component was detected to be 98.95% Cu (Figure 6c), is present. This is because the current efficiency of the hydrogen evolution reaction is not efficient enough to change the growth conditions of Cu. Figure 6d is the magnified image of area 1 in Figure 6b, it is clear that Cu grew in a dendritic form, which agrees with the data reported in [22].

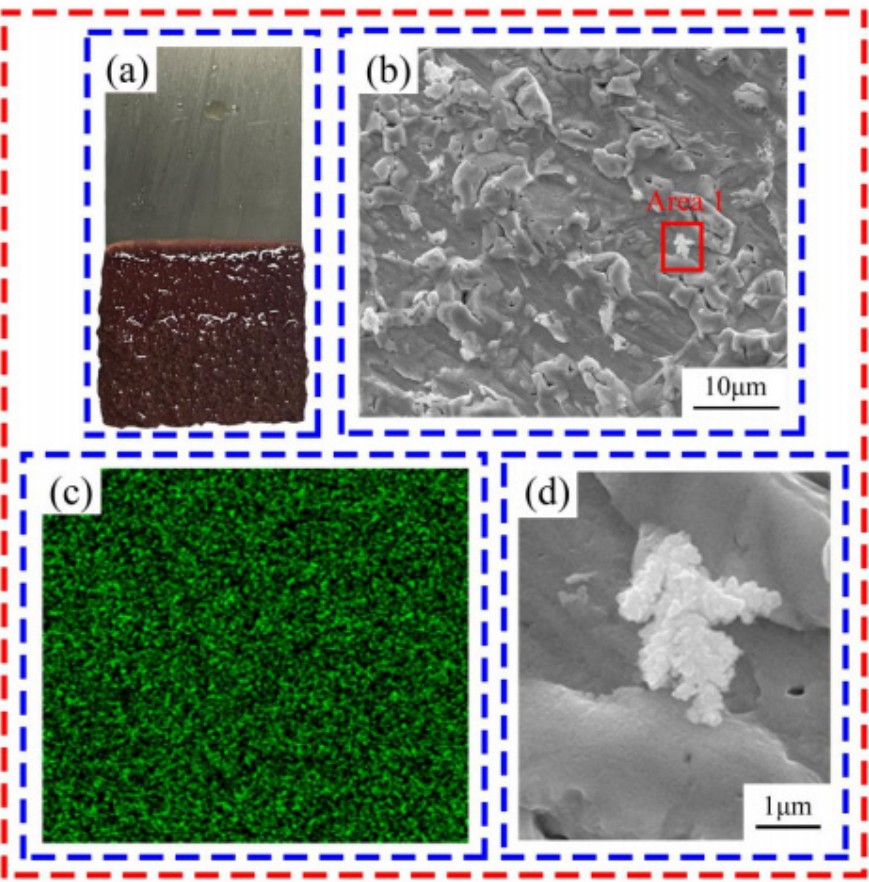

**Figure 6.** (**a**) Cathode powder, (**b**) SEM image of Cu foil, (**c**) element mapping of Cu of Figure 6b, and (**d**) the magnified image of area 1 in Figure 6b.

### 3.3. Effect of H$_2$SO$_4$ Concentration on Recovery Rate and Purity of Cu

Figure 7 demonstrates the effect of H$_2$SO$_4$ concentration on the purity and recovery rate of the metals. In this experiment, the increase in sulfuric acid concentration from 1 to 4 mol/L led to an increase in the recovery rate. The other conditions were: slurry concentration of 30 g/L, current density of 80 mA/cm$^2$, reaction time of 7 h, and reaction temperature of 15 °C. The recovery rates of Cu, Sn, and Ni in the cathode and electrolyte after the slurry electrolysis experiment are presented in Figure 7a,b. As shown in Figure 7a, the recovery rate of Cu in the cathode after the experiment increased from 65.10% to 77.77%, with the increase in H$_2$SO$_4$ from 1 to 4 mol/L. The recovery rates of Sn and Ni in the cathode increased with the H$_2$SO$_4$ concentration increasing from 3 mol/L to 4 mol/L, from 26.50% to 55.25% and 11.59% to 15.51%, respectively. Figure 7b indicates that the recovery rate of Cu in the electrolyte, after the experiment, remained at a fairly low level. At H$_2$SO$_4$ concentrations higher than 2 mol/L, the recovery rate of Sn in the electrolyte, after the experiment, rapidly increased, from 28.64% to 85.40%. The recovery rate of Ni in the electrolyte, after slurry electrolysis, remained in a certain range.

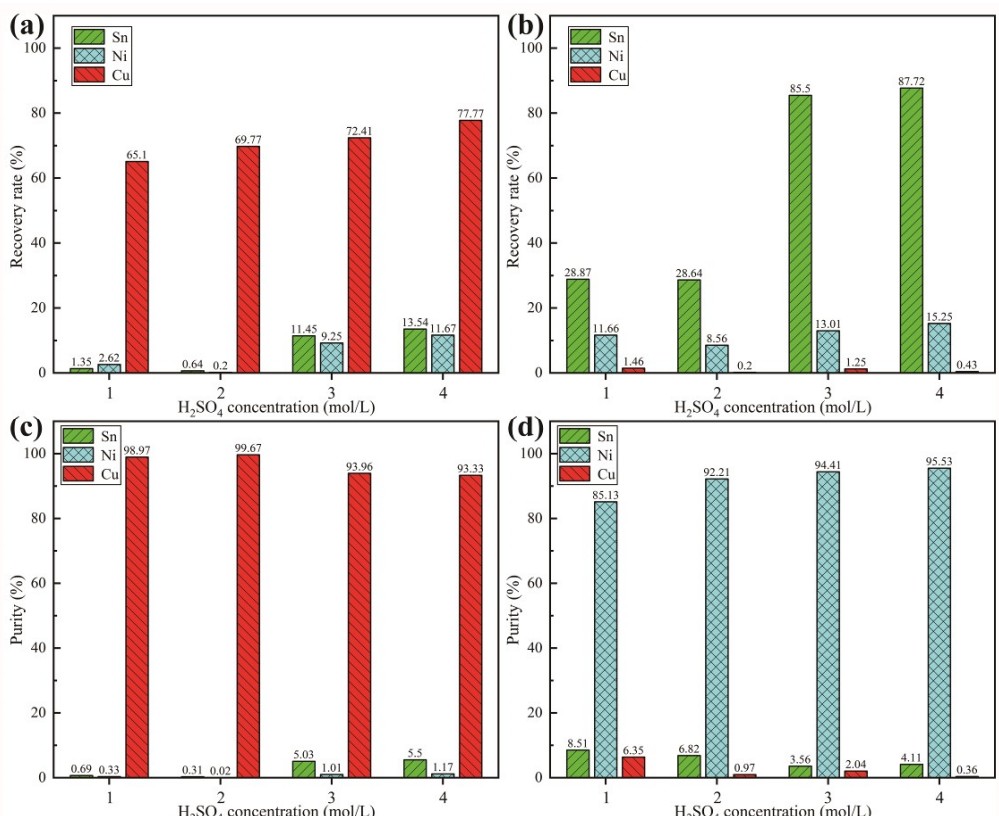

**Figure 7.** Effect of $H_2SO_4$ concentration on (**a**) recovery rate in the cathode, (**b**) recovery rate in the electrolyte, (**c**) purity in the cathode, and (**d**) purity in the electrolyte.

Figure 7c shows the purity of Cu in the cathode after the slurry electrolysis experiment, the purity of Cu first increased from 98.97% to 99.67% and then decreased to 93.33%, being highest at the $H_2SO_4$ concentration of 2 mol/L. The purity of the Sn and Ni in the cathode, after slurry electrolysis, remained at a low level as the $H_2SO_4$ concentration increased. The purity of Cu in the electrolyte after the experiment decreased, from 6.35% to 0.36%, when the $H_2SO_4$ concentration increased, as displayed in Figure 7d. The purity of Ni in the electrolyte decreased from 8.51% to 4.11% when the $H_2SO_4$ concentration increased, meanwhile, the purity of Sn in the electrolyte increased from 85.13% to 95.53%, as shown in Figure 7d.

As the $H_2SO_4$ concentration increased, not only did the recovery rate of Cu in the cathode remain stable, at near 70% when the $H_2SO_4$ concentrations were 2 and 4 mol/L, but the recovery rate of Ni and Sn increased. In the electrolyte, Cu could hardly be found, and the recovery rate of Ni remained around 10%, but the recovery rate of Sn increased. The slurry reacted more rapidly with the electrolyte when the $H_2SO_4$ concentration was higher, which caused a higher concentration of $Cu^{2+}$, $Ni^{2+}$, and $Sn^{2+}$, and more deposition of metal ions in the cathode. The increase in the concentration of $H_2SO_4$ leads to an increase in the concentration of $H^+$ in the electrolyte, which facilitates the leaching of metal from the anode, thus increasing the concentration of metal ions in the electrolyte, but the concentration of $H^+$ can lead to severe hydrogen precipitation reactions if it exceeds a certain limit. The recovery of Cu in the electrolyte is always smaller than that of Ni and Sn, because Cu has a more positive potential compared to Ni and Sn and can be deposited preferentially at the cathode. Compared to Ni and Sn, Sn has a more positive potential and can be deposited preferentially, so as the $H_2SO_4$ concentration increases, the recovery rate of Sn at the cathode grows faster than that of Ni, making the recovery rate of Sn greater than that of Ni. The recovery rate of Sn in both the cathode and electrolyte continued to increase with the increase in $H_2SO_4$ concentration, because the increase in $H_2SO_4$ concentration increased the $Sn^{2+}$ concentration, thus promoting the deposition of Sn at the cathode. The

fact that the recovery of Cu in the electrolyte remained close to 0, indicates that the rate of Cu deposition at the cathode was greater than the rate of Cu leaching at the anode, which resulted in a minimal amount of Cu in the electrolyte. Although the $H_2SO_4$ concentration had little effect on the cathodic Cu recovery, the increase in $H_2SO_4$ concentration increased the recovery of Sn and Ni from the cathode, resulting in a decrease in the cathodic Cu purity after the $H_2SO_4$ concentration reached 2 mol/L. In summary, the optimal $H_2SO_4$ concentration for cathodic recovery of Cu was 2 mol/L, when the cathodic recovery and purity of Cu reached the optimum.

### 3.4. Effect of Slurry Density on Recovery Rate and Purity of Cu

Figure 8 demonstrates the effect of slurry density on the purity and recovery rate of the metals. Experiments were conducted at different slurry densities, of 30, 50, and 70 g/L. The other conditions were: $H_2SO_4$ concentration 2 mol/L, current density 80 mA/cm², reaction time 7 h, and reaction temperature 15 °C. Figure 8a shows that the recovery rate of Cu in the cathode first increased, from 69.77% to 72.45%, and then decreased to 61.77% when the slurry density increased from 30 g/L to 70 g/L. Figure 8b shows that the recovery rate of Cu in the electrolyte was 0.10% at 30 g/L, 0.08% at 50 g/L, and 0.31% at 70 g/L, and obviously, the value is approximately 0%, thus Cu cannot be found in the electrolyte. The recovery rates of Sn and Ni in the cathode increased with the increase in slurry density. Sn increased from 0.64% to 24.66%, and Ni increased from 0.24% to 20.35%, when the slurry density increased from 30 to 70 g/L, as shown in Figure 8a. The recovery rate of Sn in the electrolyte decreased, from 28.64% to 18.56%, when the slurry density increased from 30 to 70 g/L. At the same time, the recovery rate of Ni in the electrolyte increased from 10.28% to 12.61%, as shown in Figure 8b.

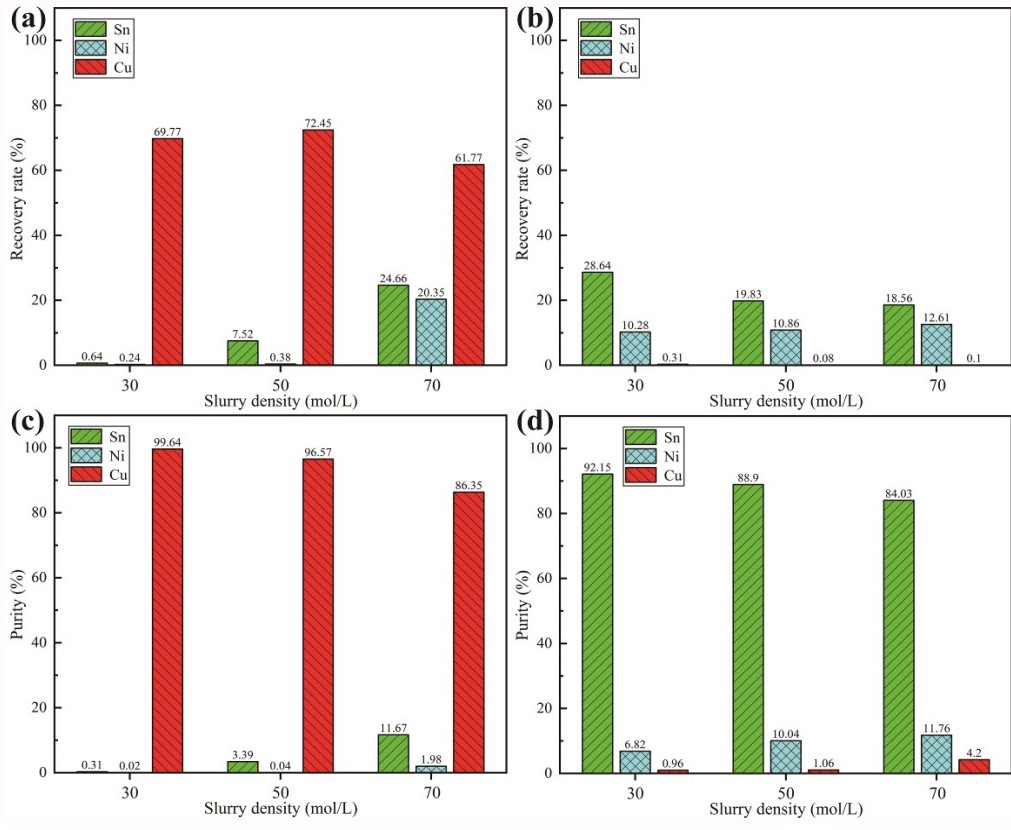

**Figure 8.** Effect of slurry density on (**a**) recovery rate in the cathode, (**b**) recovery rate in the electrolyte, (**c**) purity in the cathode, and (**d**) purity in the electrolyte.

Figure 8c illustrates that the purity of Cu in the cathode decreased from 99.64% to 86.35% when the slurry density increased from 30 to 70 g/L. Figure 8d shows that the



purity of Cu in the electrolyte increased from 0.96% to 4.20%, when the slurry density increased from 30 g/L to 70 g/L. In Figure 8c, Sn and Ni increased from 0.31% and 0.02%, to 11.67% and 1.98%, when the slurry density increased from 30g/L to 70g/L, respectively. The purity of Sn in the electrolyte decreased from 92.15% to 84.03%, when the slurry density increased from 30 to 70 g/L. Meanwhile, the purity of Ni in the electrolyte increased from 6.82% to 11.76%, as shown in Figure 8d.

As the slurry density increased, the recovery rate of Cu in the cathode kept stable, at near 70%, when the slurry density was 30 g/L and 50 g/L, and decreased at 70 g/L, while the recovery rates of Ni and Sn kept increasing. In the electrolyte, Cu could hardly be found, and the recovery rate of Ni increased, while that of Sn decreased. This was because the slurry reacted more rapidly with the electrolyte when the slurry density became higher. It caused a higher concentration of metal ions ($Cu^{2+}$, $Ni^{2+}$, $Sn^{2+}$) and the rapid deposition of metal ions ($Cu^{2+}$, $Ni^{2+}$, $Sn^{2+}$) happened in the cathode, which led to a decrease in the recovery rate of Cu.

With the increase in slurry density, the purity of Cu in the cathode decreased, and the purity of Cu in the electrolyte remained near 0%. The purity of Sn in the cathode increased, and the purity of Sn in the electrolyte decreased when the slurry density increased. The purity of Ni in the cathode was kept at a low level and the purity of Ni in the electrolyte increased, with the increase in slurry density. Because slurry with a higher density reacted with the electrolyte, causing the densities of $Cu^{2+}$, $Sn^{2+}$, and $Ni^{2+}$ to increase during electrolysis, thus speeding up ion deposition at the cathode part. The increase in slurry concentration, on the one hand, is beneficial to the leaching of scrap CPU slot metal powder in the anode, which increases the concentration of metal ions in the electrolyte and enhances the mass transfer, thus enhancing the deposition of metals in the cathode. Cu is not active enough, compared to Ni and Sn, so Ni and Sn are more easily leached in the anode. But on the other hand, the increase in slurry concentration means more metal powder of waste CPU slots in the anode chamber, which will reduce the specific surface area of the metal powder in contact with the electrolyte and thus affect the rate of metal leaching. This indicates that the recovery rate of Cu at the cathode does not increase or decrease monotonically; at the beginning of the slurry concentration increase, the metal ion concentration in the electrolyte increases, making the recovery rate of Cu increase. As the slurry concentration continues to increase, the specific surface area of the electrolyte in contact with the metal powder of the waste CPU slots becomes smaller and smaller, making the metal ion concentration within the electrolyte decrease, and no longer favorable for the leaching of metals. Since both Ni and Sn leached with higher priority than Cu, and were less affected by the decrease in specific surface area than Cu was, the amount of Ni and Sn leached in the electrolyte increased, and the purity of $Cu^{2+}$ in the electrolyte decreased, ultimately leading to a decrease in the recovery of Cu. Although the recovery of Cu at the cathode was 70.25% when the slurry concentration was 50 g/L, which was slightly higher than the recovery of Cu at 30 g/L, of 66.96%. The purity of Cu at a slurry concentration of 50 g/L was lower than that at 30 g/L. Considering that this experiment aims at cathodic recovery of Cu with higher purity, it is in accordance with the experimental purpose to have the highest possible purity under the condition that the recovery rate is considerable.

### 3.5. Effect of $NH_4Cl$ Concentration on Recovery Rate and Purity of Cu

Figure 9 indicates the effect of $NH_4Cl$ concentration on the recovery rate and purity of Cu, Sn, and Ni. Experiments were conducted at different $NH_4Cl$ concentrations of 30, 60, and 90 g/L. The other conditions were: $H_2SO_4$ concentration 2 mol/L, slurry concentration 30 g/L, current density 80 mA/cm$^2$, reaction time 7 h, and reaction temperature 15 °C. Figure 9a shows that when the $NH_4Cl$ concentration was increased from 30 to 90 g/L, the recovery rates of the metals increased: Cu increased from 62.12% to 96.19%, Sn increased from 3.67% to 14.09%, and Ni increased from 3.78% to 17.17%. Figure 9c shows that the purity of Cu in the cathode was 97.47%, 96.53%, and 93.72%, respectively, when the $NH_4Cl$ concentration was increased from 30 to 90 g/L. Meanwhile, the purity of the Sn and Ni

remained at a low level (less than 5%). In Figure 9b, when the NH$_4$Cl concentration increased from 30 to 90 g/L, the recovery rates of Cu and Sn in electrolysis decreased from 3.71% and 1.03%, to 0.85% and 0.22%, respectively, but the recovery rate of Ni was kept at a high level in electrolysis. In Figure 9d, the purity of Cu and Sn in electrolysis decreased, while that of Ni in electrolysis increased, when the NH$_4$Cl concentration increased from 30 g/L to 90 g/L.

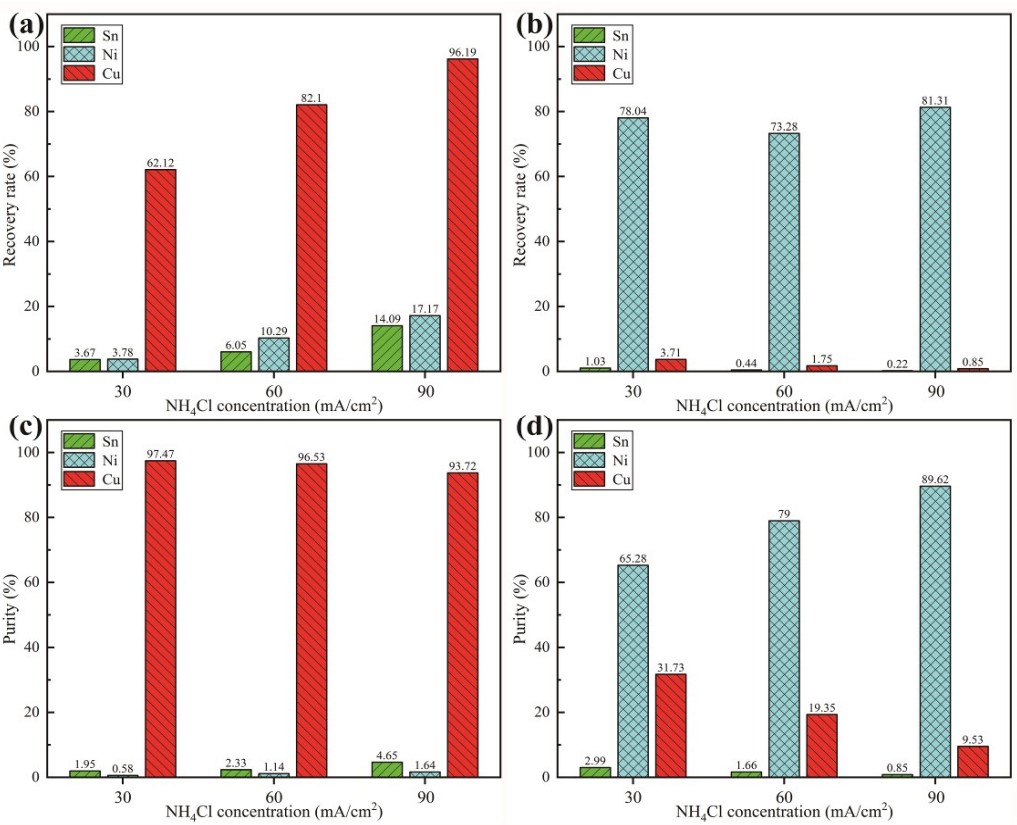

**Figure 9.** Effect of NH$_4$Cl concentration on (**a**) recovery rate in the cathode, (**b**) recovery rate in the electrolyte, (**c**) purity in the cathode, (**d**) purity in the electrolyte.

With the increase in NH$_4$Cl concentration, the recovery rate of Cu in the cathode increased to 96.19%, when the NH$_4$Cl concentration was 90 g/L. These values were rising, and the last one was close to 100%, so any increase after 90 g/L in the NH$_4$Cl concentration would not have a significant impact on the recovery rate of Cu. At the same time, the recovery rate of Sn and Ni both showed an increasing trend. The trend of the metal recovery rate shows that the increase in NH$_4$Cl concentration is beneficial to Cu recovery, reaching the highest recovery rate at 90 g/L. Because metal recovery rate increased with increasing NH$_4$Cl concentration, Cu recovery increased at a faster rate than Sn and Ni recovery. Although the purity of Cu decreased with increasing NH$_4$Cl concentration, the purity of Cu remains at a high level (over 90%). Because Cl$^-$ can form CuCl, CuCl$_2$, [CuCl$_2$]$^-$, [CuCl$_3$]$^{2-}$, and other compounds or ions with Cu, increasing the concentration of NH$_4$Cl both enhances the concentration of Cl$^-$ and facilitates the leaching of Cu from the anode. At the same time, increasing the concentration of NH$_4$Cl increases the electrical conductivity of the electrolyte and helps the dissolution of the metal, so that while Cu is dissolved, Sn and Ni are also leached, as impurities. The increase in metal concentration enhances the deposition ability of metals, so with the increase in NH$_4$Cl concentration, the recovery of all three metals at the cathode is increased, and the increase in Cu is the largest. Ni, as an active metal, can leach a lot in the electrolyte with the increase in electrolyte conductivity, but because Ni is not easily deposited in the cathode, so the electrolyte recovery of Ni is high. If the concentration of NH$_4$Cl exceeds 90 g/L, the room for improvement of Ni

recovery and Sn recovery at the cathode far exceeds that of Cu recovery, thus leading to a decrease in the purity of Cu at the cathode, so there is no need to continue to increase the concentration of NH$_4$Cl. Overall, a NH$_4$Cl concentration of 90 g/L is the optimum concentration for Cu recovery.

### 3.6. Effect of Current Density on Recovery Rate and Purity of Cu

Figure 10 displays the effect of current density on the recovery rate and purity of Cu, Sn, and Ni. Experiments were conducted at different current densities, of 40, 80, and 120 mA/cm$^2$. The other conditions were: H$_2$SO$_4$ concentration 2 mol/L, slurry concentration 30 g/L, reaction time 7 h, and reaction temperature 15 °C. Figure 10a shows that the recovery rate of Cu increased from 61.94% to 98.73%, when the current density increased from 40 mA/cm$^2$ to 120 mA/cm$^2$. The increasing current also caused the recovery rate of Ni to increase from 2.70% to 87.92%, and the recovery rate of Sn from 1.09% to 15.28%. However, the increase in the value of the recovery rate of Cu was not obvious, while the increase in the recovery rate of Ni was nearly five times higher that of Cu, with the increase in current density from 80 mA/cm$^2$ to 120 mA/cm$^2$. Figure 10c shows that the purity of Cu in the cathode was 99.51%, 93.72%, and 86.36%, when the current density was increased from 40 mA/cm$^2$ to 120 mA/cm$^2$. Meanwhile, the purity of Sn and Ni increased at low current density levels. When the current density increased from 40 to 120 mA/cm$^2$, the recovery rate of Cu and Sn in electrolysis decreased from 27.55% and 1.18%, to 0.16% and 0.20%, respectively, but the recovery rate of Ni increased firstly and then decreased (Figure 10b). Additionally, the purity of Sn in electrolysis decreased when the current density increased. The purity of Cu in electrolysis decreased first and then increased when current density increased, while the purity of Ni in electrolysis increased first and then decreased (Figure 10d).

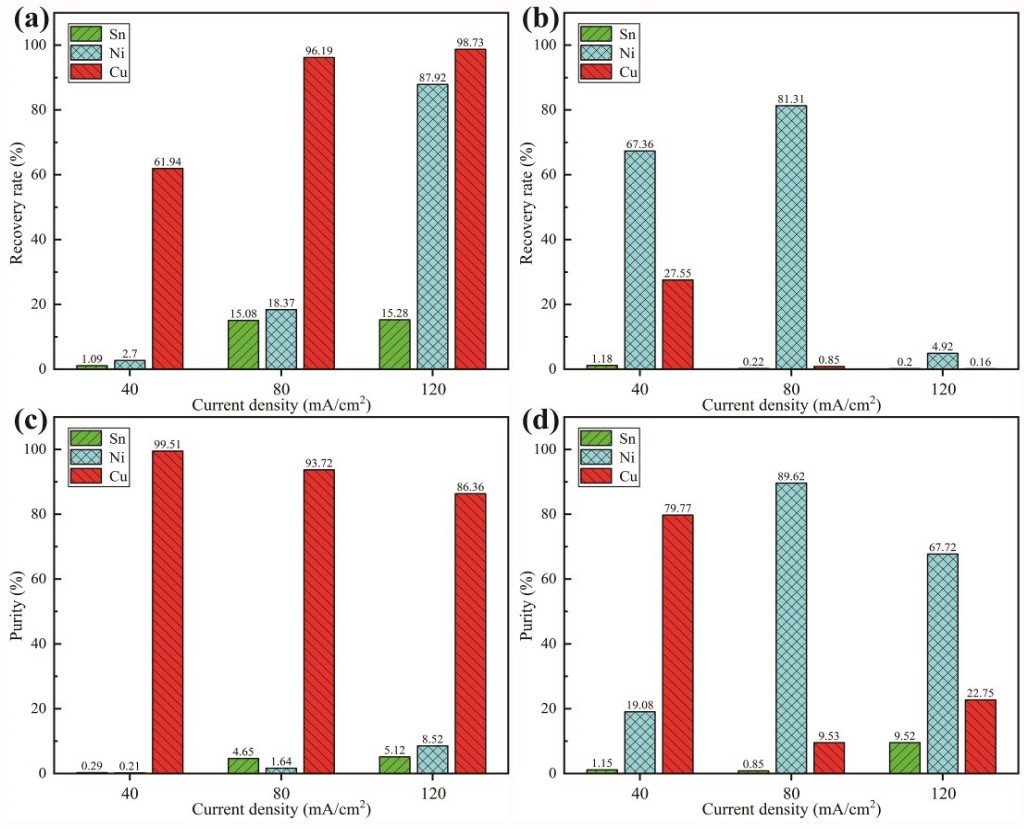

**Figure 10.** Effect of current density on (**a**) recovery rate in the cathode, (**b**) recovery rate in the electrolyte, (**c**) purity in the cathode, and (**d**) purity in the electrolyte.

With the increase in current density, the recovery rate of Cu in the cathode increased from 61.94% to 98.73%, when the current density was 120 mA/cm$^2$. These values were rising and the last one was close to 100%, so any increase after 120 mA/cm$^2$ in the current density would not have a significant impact on the recovery rate of Cu. Meanwhile, the recovery rates of Sn and Ni both showed an increasing trend, and that of Ni increased a lot when the current density increased from 80 mA/cm$^2$ to 120 mA/cm$^2$. The trend of the metal recovery rate shows that the increase in current density is beneficial to Cu recovery, reaching the highest recovery rate at 80 mA/cm$^2$. The recovery rate of Cu increased faster than that of Sn and Ni up to 80 mA/cm$^2$, but at 120 mA/cm$^2$, the recovery rate of Ni was very close to that of Cu. Moreover, the purity of Cu decreased from 40 to 120 mA/cm$^2$ of current density, the purity of Cu decreased to less than 90% when the current density was increased to 120 mA/cm$^2$. The increase in current density will enhance the mass transfer rate of metals in electrolysis, thus accelerating the cathodic deposition rate of the metals. When the current density is 40 mA/cm$^2$, a large number of metal ions are stored in the electrolyte and only some of the metal is deposited at the cathode, while the metal powder of the used CPU slot contains more Cu elements, so most of the metal ions in the electrolyte are Cu$^{2+}$. Because Cu has the advantage of depositing first at the cathode, the purity of the Cu recovered from the cathode is higher when the total metal deposition is relatively small. However, considering that there is still a large amount of Cu$^{2+}$ present in the electrolyte, increasing the current density will improve the recovery of Cu at the cathode, so when the current density is increased to 80 mA/cm$^2$, the recovery of Cu at the cathode is greatly enhanced. The increase in current density can improve the recovery of cathodic Cu, but too high a current density makes other metal ions gain enough energy to accelerate the deposition at the cathode, so when the current density reaches 120 mA/cm$^2$, the recovery of Ni and Sn at the cathode increases significantly, which reduces the purity of the Cu recovered at the cathode. In general, 80 mA/cm$^2$ of current density is the best current density for Cu recovery.

### 3.7. Effect of Reaction Time on Recovery Rate and Purity of Cu

Figure 11 exhibits the effect of reaction time on the recovery rate and purity of Cu, Sn, and Ni. Experiments were conducted at different reaction times, of 3 h, 5 h, and 7 h. The other conditions were: H$_2$SO$_4$ concentration 2 mol/L, slurry concentration 30 g/L, current density 80 mA/cm$^2$, and reaction temperature 15 °C. Figure 11a shows that when reaction time increased from 3 to 7 h, the recovery rate of metals increased, for Cu this increased from 40.88% to 96.19%, for Sn from 0.45% to 14.09%, and for Ni from 0.36% to 17.17%. Figure 11c shows that the purity of Cu in the cathode was 99.54%, 97.46%, and 93.72%, respectively, when the reaction time was increased from 3, to 5, to 7 h. Meanwhile, the purity of Sn and Ni remained at a very low level (no more than 5%). In Figure 11b, when reaction time increased from 3 to 7 h, the recovery rate of Cu and Sn during electrolysis decreased, from 38.15% and 8.94%, to 0.85% and 0.22%, respectively, but the recovery rate of Ni stayed at a high level during electrolysis. In Figure 11d, the purity of Sn increased first and then decreased, the purity of Cu decreased, while the purity of Ni increased during electrolysis when the reaction time increased from 3 h to 7 h.

With the increase in reaction time, the recovery rate of Cu at the cathode increased to 96.19%, when the reaction time was 7 h. These values were rising and the last one was close to 100%, so any increase after 7 h in the reaction time would not have a significant impact on the recovery rate of Cu. Meanwhile, the recovery rates of Sn and Ni also showed an increasing trend with reaction time. The trend of metal recovery rate shows that the increase in reaction time was beneficial to Cu recovery, reaching the highest recovery rate at 7 h. Because the metal recovery rate went up with reaction time, the speed of Cu recovery was faster than that of Sn and Ni. Although the purity of Cu decreased when increasing the reaction time from 3 h to 7 h, the purity of Cu was kept at a high level close to 100%. Increasing the reaction time had little effect on the change in Cu purity. With the increase in reaction time, on the one hand, the metal powder of the used CPU slots fully reacted

with the electrolyte and got more energy from the current, so that more metal was leached into the electrolyte, which increased the concentration of metal ions, on the other hand, the metal ions near the cathode received more electrons, which led to more deposition of metal ions in the electrolyte and reduced the concentration of metal ions. The Cu recovery of the cathode increased with the increase in the reaction time, and if the reaction time was increased past 7 h, it could further increase the Cu recovery of the cathode. However, it can be noted that the recovery of Cu in the electrolyte was close to 0 at 7 h. Based on the results of previous experiments, it can be speculated that the Ni recovery and Sn recovery of the cathode will increase after the $Cu^{2+}$ concentration in the electrolyte is close to 0. This is because the potential of $Cu^{2+}$ is more positive than that of the other metal ions, so $Cu^{2+}$ is deposited preferentially, and when the $Cu^{2+}$ concentration is low, the impurity metals will be deposited in large quantities, thus reducing the cathode recovery deposited, thus reducing the Cu purity of the cathode recovery. In short, a reaction time of 7 h is the optimum reaction time for recovering Cu.

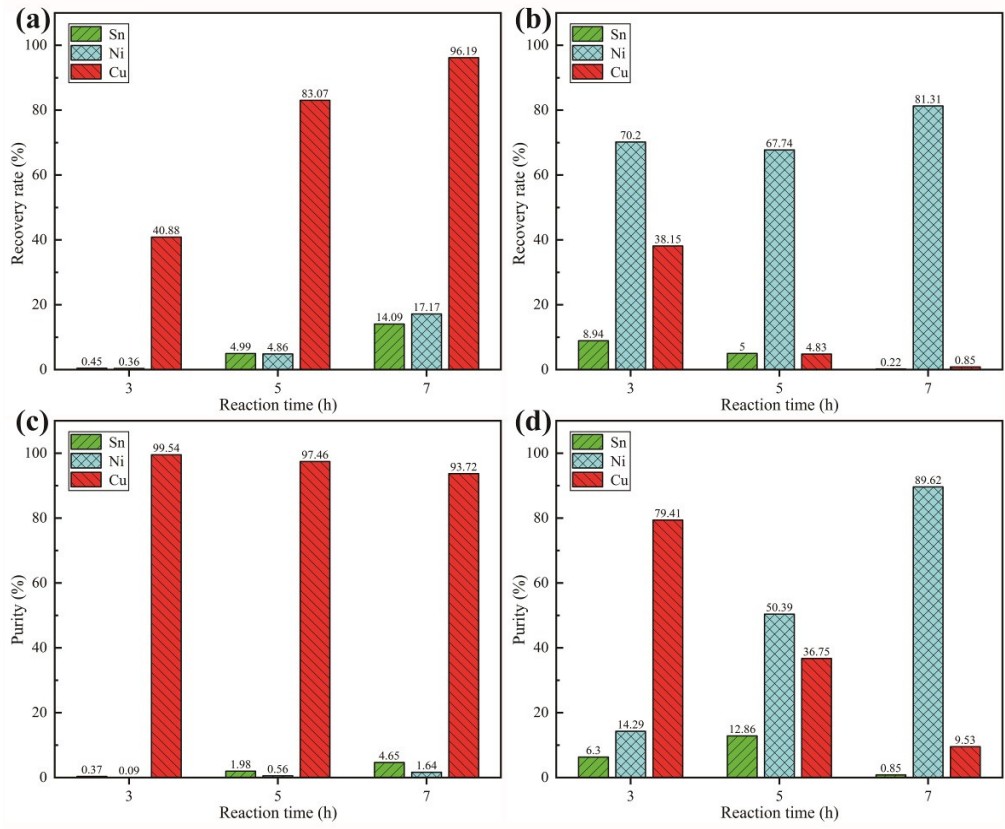

**Figure 11.** Effect of reaction time on (**a**) recovery rate in the cathode, (**b**) recovery rate in the electrolyte, (**c**) purity in the cathode, and (**d**) purity in the electrolyte.

## 4. Conclusions

In this paper, we used slurry electrolysis to recover Cu from used CPU slots, and studied the effects of different $H_2SO_4$ concentration, slurry concentration, current density, reaction time, and reaction temperature on the recovery and purity of Cu recovered from the cathode. In addition, the slurry electrolysis method was improved, to enhance the cathode Cu recovery rate, by adding $NH_4Cl$ and changing the current mode, while ensuring the Cu purity of the cathode recovery. In general, the recovery rate and purity of the Cu in the cathode were up to 96.19% and 93.72%, respectively, with the optimum conditions being when the $H_2SO_4$ concentration was 2 mol/L, the slurry density was 30 g/L, the $NH_4Cl$ concentration was 90 g/L, the current density was 80 mA/cm$^2$, and the reaction time was 7 h. Although some progress has been made in the research work on slurry electrolysis in this paper, there are still many aspects that deserve further study. For example, the

effects of various experimental parameters on the powder particle size of recovered metals, current efficiency, etc., remain to be studied. Lastly, we hope to achieve higher recovery efficiency of Cu using the simplest device and method.

**Author Contributions:** W.C.: conceptualization, methodology, sample preparation, formal analysis, investigation, data curation, writing—original draft, and visualization. M.L.: investigation, validation, methodology, and writing—review and editing. J.T.: resources and supervision. All authors have read and agreed to the published version of the manuscript.

**Funding:** This work was supported by the National Natural Science Foundation of China (No. 51864034).

**Institutional Review Board Statement:** Not applicable.

**Informed Consent Statement:** Not applicable.

**Data Availability Statement:** All data generated or analysed during this study are included in this published article.

**Conflicts of Interest:** The authors declare that they have no known competing financial interests or personal relationships that could have appeared to influence the work reported in this paper.

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
