# Peer review of "Efficient Recovery of Cu from Wasted CPU Sockets by Slurry Electrolysis"

_metals, doi:10.3390/met13040643_

Round 1

Reviewer 1 Report

Comment 1

What is the role of NH4Cl in this slurry electrolysis system?  It is not clear what reaction or process the NH4Cl is involved with, or what results from the addition of NH4Cl to the system?

Comment 2

The recovery of copper metal on the cathode is reported in this manuscript.  Is the recovered copper deposited only on the surface of the cathode, or does the copper metal precipitate as copper metal inside the solution present in the cathode compartment in addition to being deposited on the cathode surface?

Comment 3

The introduction can be improved as to the advantage of using the slurry electrolysis method for recovery of copper from e-waste (CPU sockets).

Comment 4

The name of the chemical reagents used (Section 2.1) should be corrected as follows:

nitrate (HNO3); replace nitrate with nitric acid.

sulfate (H2SO4); replace sulfate with sulfuric acid.

Comment 5

What is the pH of the system (the anode compartment, the cathode compartment) ?

Comment 6

Figure 7. Effect of H2SO4 concentration is reported. 

What are the experimental conditions for the other system parameters of NH4Cl concentration, current density, slurry density and reaction time?

Comment 7

Figure 8. Effect of slurry density is reported.

What are the experimental conditions for the other system parameters of NH4Cl concentration, current density, reaction time, and H2SO4 concentration?

Comment 8

Figure 9. Effect of NH4Cl concentration is reported.

What are the experimental conditions for the other system parameters of current density, reaction time, slurry density and H2SO4 concentration?

Comment 9

Figure 10. Effect of current density is reported.

What are the experimental conditions for the other system parameters of NH4Cl concentration, slurry density, reaction time, and H2SO4 concentration?

Comment 10

Figure 11. Effect of reaction time is reported.

What are the experimental conditions for the other system parameters of NH4Cl concentration, current density, slurry density, and H2SO4 concentration?

Author Response

Thank you very much for your approval of our manuscript. We have studied the valuable comments from you into consideration in preparing our revision at this time. The point to point responds to the comments are listed as following, which we would like to submit for your kind consideration.

Comment 1: What is the role of NH4Cl in this slurry electrolysis system? It is not clear what reaction or process the NH4Cl is involved with, or what results from the addition of NH4Cl to the system?

Response: The role of NH4Cl in this slurry electrolysis system is mainly to increase the concentration of Cl ions. Because Cl ions could form complexes with metals, and improve the solubility and efficiency of metals in the anode chamber. Meanwhile, the addition of ammonia ions does not add new metal ions which can ensure that no metal ions other than Cu are present.

Comment 2: The recovery of copper metal on the cathode is reported in this manuscript. Is the recovered copper deposited only on the surface of the cathode, or does the copper metal precipitate as copper metal inside the solution present in the cathode compartment in addition to being deposited on the cathode surface?

Response: The recovered Cu is mainly deposited on the cathode plate. A metal foil is first formed on the cathode plate and then deposited as a metal powder outside the metal foil.

Comment 3: The introduction can be improved as to the advantage of using the slurry electrolysis method for recovery of copper from e-waste (CPU sockets).

Response: Slurry electrolysis is a technique for simultaneous leaching and electrodeposition. It is very efficient, and the reaction unit is simple. Also, we have highlighted the point in the introduction as you suggested.

Comment 4: The name of the chemical reagents used (Section 2.1) should be corrected as follows: nitrate (HNO3); replace nitrate with nitric acid. sulfate (H2SO4); replace sulfate with sulfuric acid.

Response: Thank you for your suggestion, and we have corrected the misrepresentation.

Comment 5: What is the pH of the system (the anode compartment, the cathode compartment)?

Response: We apologize very much for missing this point. During the experiment, the pH of the system (anode chamber, cathode chamber) was varied, so we did not pay attention to this point in the past experiments.

Comment 6: Figure 7. Effect of H2SO4 concentration is reported. What are the experimental conditions for the other system parameters of NH4Cl concentration, current density, slurry density and reaction time?

Response: Thank you very much for your suggestion, we unified the other conditions in this section and only changed the concentration of H2SO4 for the test. This part was carried out at different H2SO4 concentrations (1 mol/L, 2 mol/L, 3 mol/L and 4 mol/L) and other conditions were: slurry concentration of 30 g/L, current density of 80 mA/cm2, reaction time of 7 h and reaction temperature of 15 ℃. And the corresponding expressions have been added to the manuscript. Similarly for the following questions.

Comment 7: Figure 8. Effect of slurry density is reported. What are the experimental conditions for the other system parameters of NH4Cl concentration, current density, reaction time, and H2SO4 concentration?

Response: At different slurry concentrations (30 g/L, 50 g/L and 70 g/L), the other conditions were: H2SO4 concentration 2 mol/L, current density 80 mA/cm2, reaction time 7 h and reaction temperature 15 °C.

Comment 8: Figure 9. Effect of NH4Cl concentration is reported. What are the experimental conditions for the other system parameters of current density, reaction time, slurry density and H2SO4 concentration?

Response: At different NH4Cl concentrations, the other conditions were: H2SO4 concentration 2 mol/L, slurry concentration 30 g/L, current density 80 mA/cm2, reaction time 7 h, and reaction temperature 15 °C.

Comment 9: Figure 10. Effect of current density is reported. What are the experimental conditions for the other system parameters of NH4Cl concentration, slurry density, reaction time, and H2SO4 concentration?

Response: At different current densities, the other conditions were: H2SO4 concentration 2 mol/L, slurry concentration 30 g/L, reaction time 7 h, and reaction temperature 15 °C.

Comment 10: Figure 11. Effect of reaction time is reported. What are the experimental conditions for the other system parameters of NH4Cl concentration, current density, slurry density, and H2SO4 concentration?

Response: At different reaction times, the other conditions were: H2SO4 concentration 2 mol/L, slurry concentration 30 g/L, current density 80 mA/cm2 and reaction temperature 15 °C.

Reviewer 2 Report

Submitted paper is written fine, however some improvements should be made in order to continue its publishing process. Review report is given in the attached file, please consider all the comments and suggestion provided within.

Author Response

Thank you very much for your approval of our manuscript. We have studied the valuable comments from you into consideration in preparing our revision at this time. The point to point responds to the comments are listed as following, which we would like to submit for your kind consideration.

Comment 1:

Abstract:

Abstract should be improved and extended since it does not have all elements every abstract should have.

Response: We appreciate it very much for this good suggestion, and we have done it

according to your idea.

Comment 2:

Introduction:

  1. Some improvements could be made. Such as: https://doi.org/10.3390/met12122021, https://doi.org/10.1007/s11837-004-0237-9, https://doi.org/10.1007/BF03403310.
  2. Line 33-35: This sentence should be reconstructing.
  3. Line 38: Please, reconstruct this sentence in the way it states facts.
  4. Line 72: Please, put into brackets the meaning of acronym MLCC.
  5. Line 78-79: Please correct the definition of acronym PCB.

Response: Thank you for your careful review and approval, we trust that these comments will be helpful to our work. With the changes using your comments, the new introduction is shown in the revised manuscript.

Comment 3:

Materials and methods

  1. Some improvements can be made as well the most common mistake is in grammar.
  2. Line 89-91: It is recommended to mention the manufacturers of the chemicals used.
  3. Line 97: Please insert the name of the CPU socket model.
  4. Line 100: Please insert the name of the cutting mill manufacturer and not only the model name, also it is recommended to describe operation parameters setup used to obtain such grade.
  5. Line 107: Notice that XRD is used to determine mineralogy composition of powdered sample.
  6. Line 120: Conjunction and is used twice, correct this.
  7. Line 122: Name the DC supplier manufacturer and model.
  8. Line 123-125: Please reconstruct the sentence, it could be better to say:” … while the weight of NH4Cl is varied (x-y grams).”
  9. Line 125-131: Please rephrase these sentences and make them more concise.

Response: Thank you again for your suggestions. In this section, our description is not detailed enough. Therefore, we have supplemented the content. We will answer your questions about the unmodifiable issues one by one here.

Q4: We bought the crusher from a Taobao store called "Experimental Instruments and Equipment Store". They also do not have a clear manufacturer. I am very sorry. In addition, his operation is very simple, first is to power on, then is to turn on the crushing switch, select the crushing level (strong, medium, weak), and finally start crushing, the time is their own control of the power on and off settings. That's all.

Comment 4:

Results and discussion

  1. Line 166: It is not appropriate to use word “meanwhile” instead use the word “also”.
  2. Line 170: Please double check all the reflections with those readily present in JCPDS database and check only the reflections that would be connect to mineral phase with high certainty.
  3. Line 172: Please provide the photo with the EDS element mapping of all three elements on every single microphotography.
  4. Line 189: In the abstract it is mentioned that purity of obtained powder is 93.72 % copper. Which value is right since it is written that copper purity in the experiments is up to 93.72 %?
  5. Line 194-195: In figure title should be mentioned the process parameters when such sample (cathode powder is obtained), the current density, concentration of the acid and concentration of the NH4Cl.
  6. Line 203: Word concentration is missing: “… the increase of sulfuric acid concentration from 1 to 4 mol/L lead to an increase in recovery rate.” The sentence should be written in this manner.
  7. Line 207: What is the value of nickel recovery rate? Please insert this parameter into the sentence.
  8. Line 216: Explain the results from Figure 7. Everyone can see from the figures the recovery rates and purities of the metals, but discussion of the results is required. Please do this throughout all the subsections from 3.3 to 3.7. In addition, it can be noticed that designations for recovery rate and purity given by equations 5 and 6 (see the lines 156 and 160) are not used in the manuscript, what is the reason for this?
  9. Line 227: See the previous comment, this section lack of discussion (not the observation of the results but actual scientific discussion). Do the same for all other subsections, lines 267, 295 and 329.

Response: We appreciate it very much for this good suggestion, and we have done it

according to your idea. Some of the questions that need to be answered are as follows:

Q2: Thank you for pointing out this error, I had this same question when analyzing the XRD pattern. Their composition is complex since we could not find a card in the JCPDS database that corresponds highly to it. Moreover, for these several elements, some of their characteristic peaks overlap, so the process of labeling the characteristic peaks is difficult and it is possible to miss them. We currently only indicate the characteristic peaks by comparing the cards of the individual elements, while combining the number of theoretical components of the element as a priority for labeling.

Q3: Thank you for your suggestions regarding EDS. In our opinion, this image is also unattractive because it seems incomplete. In fact, it is complete and the result of all the experiments. As we described, the elements in the CPU socket were different in each location, and EDS was unable to scan for elements that were not present, even if we forced the element to be added, EDS provided us with an evenly distributed picture of the "snowflake dots". No matter what element you force to add, its outline is evenly distributed "snowflake dots". So, even though this image is not very pretty, we have expressed it this way. We hope you will understand.

Q4: The Cu purity of the metal foil is high, and there is a layer of metal powder outside the metal foil Cu concentration is slightly reduced.

Q8&Q9: Thank you very much for pointing out the shortcomings to us. In the revised manuscript, we have added the relevant discussion. In addition, regarding the cases where the corresponding formula names were not used, we made a modification to add an abbreviation note to the formula description.

Comment 5:

Conclusion

This section must be improved since it contains one short paragraph of everything that is mentioned in the section Results and Discussion.

Response: Thank you for your valuable comments about the conclusion, which have greatly improved the level of our articles. We have made a large number of revisions in the revised manuscript to enhance the conclusion of the article.

Reviewer 3 Report

The authors discussed the effectiveness of slurry electrolysis for the recovery of base metals from electronic waste. Electronic wastes contain a significant amount of copper and efficient copper recovery from similar secondary sources can improve environmental sustainability. Slurry electrolysis is a fast and effective method of copper recovery.

The authors studied the effects of critical parameters that influence the slurry electrolysis experiment. The choice of parameters is extensive and covers all aspects.

However, the authors undertook a one-factor-at-a-time method to determine the optimum condition. However, this method fails to encompass the synergistic effect among the process parameters. Additionally, the number of experiments actually require to optimize the system is very large. It is advised to carry out statistical-based optimization methods such as the Taguchi method or response surface optimization methods for efficient determination of process optimization.   

Figures 7-11 in the article, require error bars and authors are requested to input the information on the number of independent experiments carried out in determining the experimental values. 

Author Response

Thank you very much for your approval of our manuscript. We have studied the valuable comments from you into consideration in preparing our revision at this time. The point to point responds to the comments are listed as following, which we would like to submit for your kind consideration.

Comment: The authors discussed the effectiveness of slurry electrolysis for the recovery of base metals from electronic waste. Electronic wastes contain a significant amount of copper and efficient copper recovery from similar secondary sources can improve environmental sustainability. Slurry electrolysis is a fast and effective method of copper recovery.

The authors studied the effects of critical parameters that influence the slurry electrolysis experiment. The choice of parameters is extensive and covers all aspects.

However, the authors undertook a one-factor-at-a-time method to determine the optimum condition. However, this method fails to encompass the synergistic effect among the process parameters. Additionally, the number of experiments actually require to optimize the system is very large. It is advised to carry out statistical-based optimization methods such as the Taguchi method or response surface optimization methods for efficient determination of process optimization.

Figures 7-11 in the article, require error bars and authors are requested to input the information on the number of independent experiments carried out in determining the experimental values.

Response: We appreciate it very much for this good suggestion, and we have done our best to revise the manuscript according to your ideas. We apologize that we did not perform multiple experiments and statistical analyses due to experimental cost constraints. So each part of the experiment was analyzed by one EDS scan to obtain the component ratios. Although it was not possible to perform a large number of experiments again, we added the accuracy of EDS scan used in the experiments to the manuscript as follows: " The relative error of EDS analysis varies with the elemental content; the smaller the content, the larger the relative error. The relative error is 2% when the content is >20wt.%, 10% when the content is between 3wt.%-20wt.%, 30% when the content is between 1wt.%-3wt.%, and 50% when the content is <1wt.%."

Round 2

Reviewer 1 Report

The authors have adequately improved their manuscript.